# Advancing Adverse Drug Reaction Prediction with Deep Chemical Language Model for Drug Safety Evaluation

**DOI:** 10.3390/ijms25084516

**Published:** 2024-04-20

**Authors:** Jinzhu Lin, Yujie He, Chengxiang Ru, Wulin Long, Menglong Li, Zhining Wen

**Affiliations:** 1College of Chemistry, Sichuan University, Chengdu 610064, China; 2Medical Big Data Center, Sichuan University, Chengdu 610064, China

**Keywords:** adverse drug reactions, drug safety evaluation, deep chemical language model, structural alerts, deep learning

## Abstract

The accurate prediction of adverse drug reactions (ADRs) is essential for comprehensive drug safety evaluation. Pre-trained deep chemical language models have emerged as powerful tools capable of automatically learning molecular structural features from large-scale datasets, showing promising capabilities for the downstream prediction of molecular properties. However, the performance of pre-trained chemical language models in predicting ADRs, especially idiosyncratic ADRs induced by marketed drugs, remains largely unexplored. In this study, we propose MoLFormer-XL, a pre-trained model for encoding molecular features from canonical SMILES, in conjunction with a CNN-based model to predict drug-induced QT interval prolongation (DIQT), drug-induced teratogenicity (DIT), and drug-induced rhabdomyolysis (DIR). Our results demonstrate that the proposed model outperforms conventional models applied in previous studies for predicting DIQT, DIT, and DIR. Notably, an analysis of the learned linear attention maps highlights amines, alcohol, ethers, and aromatic halogen compounds as strongly associated with the three types of ADRs. These findings hold promise for enhancing drug discovery pipelines and reducing the drug attrition rate due to safety concerns.

## 1. Introduction

Market drugs frequently lead to unexpected adverse drug reactions (ADRs) during clinical use, presenting significant challenges to patient safety [1,2]. Among these, idiosyncratic ADRs, such as drug-induced rhabdomyolysis, pose a particular risk due to their life-threatening nature and intricate pathogenesis, compounded by their low incidence rates [3,4,5]. Despite efforts to identify clinical biomarkers and monitoring modalities for timely diagnosis during medication intake, the complexity of these ADRs and substantial inter-individual variability among patients impede the diagnostic efficacy of these clinical parameters [1,6,7]. Consequently, the accurate prediction of the risk associated with serious ADRs induced by marketed drugs becomes imperative in drug safety evaluation. Such predictions not only facilitate the prompt adjustment of medication strategies to ensure patient safety and efficacy but also serve as early detection mechanisms within the drug discovery process. Structure–Activity Relationship (SAR) methodologies, which are based on the chemical structures and properties of compounds, utilize various molecular descriptors as features and leverage machine-learning techniques to predict or interpret the relationship between the chemical structure of compounds and its biological activity or toxicological characteristics [8]. These methodologies offer pivotal approaches to correlate the chemical structure of drugs with their propensity for ADRs, representing indispensable tools for predicting drug-induced adverse reactions [9,10]. Previous research has demonstrated the utility of SAR models employing conventional machine-learning methods, including Support Vector Machine (SVM) [11,12,13,14], Random Forest (RF) [15,16,17], and Decision Forest (DF) [18,19]. Notably, these models have exhibited promising performance in predicting liver toxicity [15,18,19,20], cardiotoxicity [13,14,17,21], and other adverse reactions. In our previous investigation, we constructed datasets encompassing three distinct ADRs based on FDA-approved drug labeling information: the drug-induced QT prolongation atlas (DIQTA) [22], drug-induced teratogenicity dataset (DITD) [12], and drug-induced rhabdomyolysis atlas (DIRA) [23]. Leveraging conventional machine-learning algorithms in conjunction with molecular descriptors, we predicted the risk associated with these three types of ADRs [12,13,16].

Conventional machine-learning models, which utilize various types of molecular descriptors as features, have exhibited limitations in predicting adverse drug reactions (ADRs) [24,25]. The selection of molecular descriptors for prediction tends to be somewhat arbitrary, typically relying on experiential knowledge or the incorporation of as many descriptors as feasible, followed by an assessment of feature importance. Despite these efforts, effectively translating these descriptors into structural alerts (SAs) for analyzing structural features closely associated with ADRs during mechanistic interpretation remains challenging. Furthermore, the reliability of these models may be compromised by limited sample sizes, particularly for drugs that induce idiosyncratic ADRs. In recent years, several studies have investigated the development of chemical language models employing deep learning algorithms [26,27,28,29,30]. These models can be trained on extensive datasets of molecules to learn structural information from molecular representations, such as the Simplified Molecular Input Line Entry System (SMILES), and subsequently establish pre-trained models for predicting molecular properties. Li et al. introduced SMILES Pair Encoding (SPE) [31], which learns the vocabulary of high-frequency SMILES substrings from large chemical datasets and then tokenizes SMILES. This approach has demonstrated promise in molecular generation and Quantitative Structure–Activity Relationship (QSAR) prediction tasks. Wu et al. employed the BiLSTM (Bidirectional Long Short-Term Memory) attention network to extract key features from SMILES strings [32], achieving superior results across eleven tasks related to absorption, distribution, toxicity, and others. Ucak et al. proposed the atom in the SMILES tokenization scheme [33], which incorporates environmental information and resolves ambiguities in the general nature of SMILES tokens. These studies utilize deep learning algorithms to decipher the intricate chemical information concealed within SMILES and underscore the significant impact of appropriate tokenization on multiple chemical translation and molecular property prediction tasks. However, the performance of pre-trained chemical language models combined with deep learning methods in predicting ADRs, particularly idiosyncratic ADRs induced by marketed drugs, remains largely unexplored.

In this study, we employ MoLFormer-XL [34], a pre-trained model trained on large-scale molecular datasets, to encode molecules. This is coupled with convolutional neural networks (CNNs) to predict the risk of drug-induced QT prolongation (DIQT), drug-induced teratogenicity (DIT), and drug-induced rhabdomyolysis (DIR), exploring the performance of deep chemical language models in predicting marketed drug-induced adverse reactions. We compare the results with conventional machine-learning methods employed in previous studies to evaluate the effectiveness of the model. Furthermore, by scrutinizing the structures emphasized in the linear attention maps of drugs associated with high-risk concerns, we discerned structural alerts, such as amines, alcohols, ethers, and aromatic halogens, which may correlate with ADRs. In summary, our investigation underscores that deep chemical language models effectively capture molecular structural information, proving instrumental in predicting ADRs induced by marketed drugs. This presents a viable alternative approach to bolstering drug safety evaluation and facilitating the early detection of ADRs throughout the drug development process.

## 2. Results

### 2.1. Model Performance on Three Curated Datasets

Three meticulously curated datasets, namely DIQTA, DITD, and DIRA, were employed to assess the performance of our proposed model, MoLFormer-XL-CNN. This model combines a pre-trained deep chemical language model with CNN to predict ADRs induced by marketed drugs. Performance metrics, including accuracy, recall rate, precision, Matthew’s correlation coefficient (MCC), balanced accuracy score (BACC), F1 score, the area under the receiver operating characteristic curve (AUROC), the area under the precision–recall curve (AUPRC), and specificity, achieved by our model in predicting the risk of drug-induced QT prolongation, teratogenicity, and rhabdomyolysis, are listed in Table 1. The MCCs and recall rates achieved on the three datasets are over 0.50 and 0.85, respectively, indicating satisfactory performance in predicting these ADRs. Particularly noteworthy are the high recall rates of 0.942 and 0.974 achieved on the DIQTA and DIRA datasets, respectively, illustrating the model’s effectiveness in identifying positive samples at risk of inducing QT interval prolongation and rhabdomyolysis. However, it is noteworthy that the specificity for the DIRA dataset is 0.432, indicating a higher false-positive rate in predicting drug-induced rhabdomyolysis compared to the other two ADRs. We compared the performance of our proposed MoLFormer-XL-CNN model with state-of-the-art methods reported in previous studies (Figure 1). In previous studies, Wulin Long et al. indicated that SVM achieved the highest MCC value and recall rate (mean MCC = 0.591, mean recall rate = 0.870), predicting the high risk of inducing QT interval prolongation of marketed drugs [13]. Liyuan Kang et al. indicated that SVM performed better in detecting the marketed drugs with high teratogenic risk (mean MCC = 0.312) [12]. Yifan Zhou et al. proposed that the RF model performed the best (mean MCC = 0.46) in predicting the DIR severity of the marketed drugs [16]. The three previous studies all leveraged conventional machine-learning algorithms in conjunction with molecular descriptors. The results demonstrated the superior performance of our model in terms of both MCCs and recall rates, highlighting the effectiveness of leveraging pre-trained deep chemical language models for ADR prediction tasks. Significant improvements in MCCs, especially for the DITD dataset, were observed where the MCC increased from 0.312 to 0.503. Additionally, our model exhibited improved recall rates compared to the supervised baselines. These results demonstrate that MoLFormer-XL effectively captures the molecular features of drugs, thereby facilitating subsequent prediction tasks.

### 2.2. Attention Analysis for Drugs with a High Risk of DIQT

MoLFormer-XL’s average-pooled attention metrics encode chemical information within a molecule during the encoding of canonical SMILES, enabling the capture of spatial relations between atoms. Higher values in the average-pooled attention metrics indicate the importance of atoms or groups of atoms (structural fractions) in the prediction task. By analyzing the average-pooled attention matrices, we extracted structural information considered important by the predictive model for ADR predictions. Table 2 lists five drugs identified as high risk for inducing QT interval prolongation by the Comprehensive in vitro Proarrhythmia Assay (CiPA) initiative [35]. The structural subunits with high attention values in the MoLFormer-XL framework were circled and compared with established structural alerts (SAs) obtained from ToxAlert in the Online Chemical Database (OCHEM, https://ochem.eu/home/show.do, accessed on 6 February 2024) [36,37]. Annotations with complete attention maps are provided in Appendix A. The table reveals that MoLFormer-XL was used to report SAs associated with DIQT in previous studies, such as amines, ethers, and aromatic halogens [13]. Furthermore, Aniline structures, such as pyridine in disopyramide phosphate and the benzimidazole ring in quinidine gluconate and vandetanib, were also identified as crucial structural features associated with DIQT [38,39,40].

### 2.3. Attention Analysis for Antiepileptic Drugs with a High Risk of DIT

Currently, teratogenic risk evaluation for antiepileptic drugs primarily relies on cohort studies from pregnancy registries, comparing the incidence and risk of congenital malformations among offspring exposed to different antiepileptic drugs. Reports from three major pregnancy registries have indicated that valproate and phenytoin are associated with a high risk of teratogenicity [41,42,43,44]. Table 3 lists the circled structures of valproate and phenytoin attended by MoLFormer-XL. Annotations with complete attention maps are provided in Appendix A. For phenytoin, the unsubstituted imine nitrogen and ethane-1,1-diyldibenzene in the chemical structure were considered crucial substructures associated with teratogenicity by the model [45,46]. In the chemical structure of valproate, the carboxylic acids were highlighted and associated with teratogenicity, which is consistent with the SAs revealed in previous studies [45,47].

### 2.4. Attention Analysis for Statins

DIR is an idiosyncratic ADR, and its specific pathogenesis is still under exploration. It has been widely reported that drugs in certain therapeutic categories, such as statins, are more likely to induce rhabdomyolysis [48,49]. In the DIRA dataset, all statin drugs were categorized into the highest DIR concern level. Table 4 lists seven statin drugs in the DIRA dataset with attended substructures and reported SAs. Annotations with complete attention maps are provided in Appendix A. The table reveals that nucleophilic groups in the chemical structures of each drug, particularly alcohols and amines, were attended by MoLFormer-XL. These functional groups can be converted into highly reactive electrophilic metabolites, which can covalently bind to nucleophilic sites of biological macromolecules, such as DNA, RNA, or proteins in cells, potentially leading to cell damage and other toxic reactions [39,45,50]. Additionally, aromatic halogens [51] in the chemical structures of fluvastatin, atorvastatin, rosuvastatin, and pitavastatin were also identified as contributors to inducing rhabdomyolysis.

## 3. Discussion

Conventional QSAR methods predominantly rely on molecular descriptors, such as physicochemical properties and structural fragments, to characterize chemical compounds and correlate them with biological activities. However, deep chemical language models represent a paradigmatic shift in ADR prediction by directly encoding molecular structures from textual representations, such as SMILES strings. These models offer several advantages over conventional QSAR models. Firstly, deep chemical language models provide a more comprehensive representation of molecular structures, capturing intricate structural features inherent in SMILES strings. By encoding the entire molecular structure, including bond connectivity and spatial arrangement, these models can potentially capture subtle but important nuances that traditional descriptors might overlook. This comprehensive representation enhances the model’s ability to discern complex relationships between the molecular structure and ADRs. Secondly, deep chemical language models benefit from pre-training on large-scale molecular datasets. Leveraging extensive chemical data, these models can learn universal representations of molecular structures. Pre-training allows the model to capture generalizable patterns and features across diverse chemical compounds, facilitating transfer learning for downstream tasks such as ADR prediction. Furthermore, deep chemical language models offer flexibility and scalability in model architecture. Unlike QSAR methods, which typically rely on handcrafted features and fixed model architectures, deep chemical language models can adapt their architectures to the complexity of prediction tasks.

In this study, we introduced the MoLFormer-XL-CNN model, which combines the deep chemical language model and CNNs for predicting drug-induced adverse reactions. We utilized MoLFormer-XL to encode molecular structures, leveraging its ability to capture intricate structural features. Subsequently, we employed CNNs to predict the risk of adverse reactions. To evaluate the performance of our model, we conducted predictions on three distinct datasets: DIQTA, DITD, and DIRA. Our results demonstrated significant improvements in predictive performance compared to conventional QSAR models in previous studies. Specifically, the MoLFormer-XL-CNN model achieved MCCs of 0.702, 0.503, and 0.535 on the DIQTA, DITD, and DIRA datasets, respectively. These MCC values represent substantial enhancements over the MCCs obtained by conventional QSAR methods in previous studies (MCCs = 0.591, 0.31, and 0.46, respectively) (Figure 1). Notably, the attention mechanism embedded within the MoLFormer-XL framework provided valuable insights into the chemical substructures associated with adverse reactions. By comparing these identified substructures with established SAs, we confirmed their relevance to specific adverse reactions. For instance, the model’s attention to amines, ethers, and aromatic halogens corresponded to known SAs for drug-induced QT prolongation. Furthermore, our model identified additional substructures, such as Aniline structures, which may serve as novel SAs for adverse reactions. For drugs with teratogenic or rhabdomyolysis risks, MoLFormer-XL also highlighted similar SAs identified in previous studies. For instance, imine nitrogen and ethane-1,1-diyldibenzene were associated with teratogenicity, while nucleophilic groups in chemical structures were linked to rhabdomyolysis.

Additionally, there are several caveats worth further discussion. Firstly, the stability of the MoLFormer-XL-CNN model varied across datasets, as indicated by the standard deviations of MCC and recall rates (Figure 1). Due to the small sample size of the datasets, we applied data augmentation techniques to improve the model’s performance. Secondly, while deep chemical language models can identify important substructures, further research is needed to elucidate the specific associations between individual or combinations of substructures and the mechanisms underlying adverse reactions. Thirdly, the severity stratification of drugs in three distinct datasets is mainly based on the U.S. FDA-approved drug labeling information. If drug labeling from other countries, such as European Union (EU) countries, is used, differences in stratification may lead to differences in prediction results and attended chemical substructures.

## 4. Materials and Methods

### 4.1. Study Design

In this study, we aim to investigate the effectiveness of pre-trained deep chemical language models in predicting adverse drug reactions (ADRs) induced by marketed drugs. MoLFormer-XL, a pre-trained model with an efficient linear attention mechanism, was trained on a comprehensive dataset comprising 1.1 billion molecules. To construct the predictive model, we employed MoLFormer-XL to encode canonical SMILES representations of drugs, in conjunction with convolutional neural networks (CNNs), to predict the occurrence of drug-induced QT prolongation (DIQT), drug-induced teratogenicity (DIT), and drug-induced rhabdomyolysis (DIR). We evaluated the performance of our model against previous studies and analyzed the average-pooled attention metrics within the MoLFormer-XL framework. These metrics were instrumental in identifying chemical substructures associated with the ADRs. By extracting the attended substructures from the attention metrics and comparing them with reported structural alerts (SAs), we elucidate the model’s ability to discern relationships between molecular structures and ADR propensity. An overview of our methodology is depicted in Figure 2.

### 4.2. Data Preparation

We utilized three meticulously curated datasets as follows: the drug-induced QT prolongation atlas (DIQTA, https://www.adratlas.com/DIQTA/download accessed on 6 February 2024) [22], the drug-induced teratogenicity dataset (DITD, https://www.frontiersin.org/articles/10.3389/fphar.2022.747935/full#supplementary-material accessed on 6 February 2024) [12], and the drug-induced rhabdomyolysis atlas (DIRA, https://www.adratlas.com/DIRA/download accessed on 6 February 2024) [23] for model construction and validation. All drugs in these datasets were classified into different concern levels based on the severity outlined in their FDA-approved drug labeling.

Within the DIQTA dataset, QT-prolonging drugs were extracted by full-text searching of the FDA-approved drug labeling with QT prolongation-related keywords. The severity score was assigned based on predefined keywords, a priori knowledge, and DIQT concerns, which were determined based on the severity scores. Marketed drugs were divided into four concern levels: the most DIQT concerns, moderate DIQT concerns, ambiguous levels, and no DIQT concerns. We selected 166 drugs from the most DIQT and moderate DIQT concern levels as positive samples and 100 drugs from the no-DIQT concern level as negative samples. Subsequently, we retrieved the SMILES from the DrugBank database via the link (https://go.drugbank.com/releases/latest accessed on 6 February 2024, release version 5.1.10) [52] using the DrugBank ID. The canonical SMILES were then generated using RDKit packages in Python. Ultimately, we obtained canonical SMILES for a total of 255 drugs, comprising 156 positive samples and 99 negative samples.

In the DITD dataset, a drug was categorized into high teratogenic risk if it was indicated in drug labeling that adequate and well-controlled studies or animal studies had shown a teratogenic risk to the fetus, while a drug was categorized into low teratogenic risk if both animal and human studies showed no risk to the fetus. Then, drugs were categorized into high teratogenic risk and low teratogenic risk groups based on the FDA’s pregnancy medication classification. We utilized 67 drugs from the high teratogenic risk group as positive samples and 45 drugs from the low teratogenic risk group as negative samples. Similar to the DIQTA dataset, SMILES were obtained from the DrugBank database and converted to canonical SMILES using RDKit packages in Python. Consequently, we acquired canonical SMILES for a total of 112 drugs, including 67 positive samples and 45 negative samples.

Within the DIRA dataset, DIR drugs were extracted by full-text searching of the FDA-approved drug labeling with the keyword ‘rhabdomyolysis’. The severity score was assigned based on predefined keywords and a priori knowledge, and DIR concerns were determined based on the severity scores. Marketed drugs were classified into four concern levels: the most DIR concerns, moderate DIR concerns, less DIR concerns, and no DIR concern drugs. We designated 173 drugs with DIR concerns as positive samples and 40 no-DIR concern drugs as negative samples. Similarly, SMILES was retrieved from the DrugBank database and converted to canonical SMILES using RDKit packages in Python. Finally, we obtained canonical SMILES for a total of 194 drugs, comprising 155 positive samples and 39 negative samples.

### 4.3. Model Construction

MoLFormer is an efficient transformer encoder model that utilizes rotary positional embeddings and the linear attention mechanism. Among various MoLFormer variants, MoLFormer-XL exhibited superior performance. Constructed by training on 1.1 billion unlabeled molecules from the PubChem and ZINC datasets, MoLFormer-XL aims to learn meaningful and universal representations of chemical molecules from large-scale chemical representation data, subsequently fine-tuning for various downstream molecular property prediction tasks. In our study, we leveraged the pre-trained MoLFormer-XL to encode molecules, enhancing the capture of molecular features of marketed drugs. Subsequently, we employed CNN for further prediction tasks.

MoLFormer-XL encoded each canonical SMILES, with the resulting embedding serving as an input to the CNN model. Our CNN framework comprises two convolutional layers and two fully connected layers. The first convolutional layer expands the dimensionality of the embedding threefold to capture richer information, utilizing a kernel size of 3 × 1 and padding of 1 to maintain the spatial dimensionality of features. The second convolutional layer compresses the feature space back to the original embedding dimension for subsequent fully connected operations, with kernel size and padding remaining unchanged. Both convolutional layers incorporate batch normalization, aiming to re-learn the original encoded features at a deeper level, thereby enhancing subsequent classification and predictive accuracy. In the two fully connected layers, we introduced a dropout layer to randomly skip some neurons to prevent overfitting. Functions for optimizer, loss, activation, and output were separately set to AdamW, CrossEntropyLoss, ReLU, and softmax.

To facilitate a comparison with the predictive outcomes in previous studies, our model’s construction employed fivefold cross-validation. To enhance the model’s learning capacity during training, we utilized the SMILES enumeration method [53], which generates multiple valid representations for chemical structures from a single SMILES string. This approach enriches the diversity of structural representations and has been shown in prior studies to assist in improving the accuracy of the model’s predictions for molecular properties. The Python codes for the MoLFormer-XL-CNN model were uploaded to GitHub (https://github.com/LiSH7450/MoLFormer-XL-CNN_model accessed on 6 February 2024).

### 4.4. Evaluation of Model Performance

To evaluate the performance of the model, we used nine performance metrics in this study, including accuracy, recall rate, precision, Matthew’s correlation coefficient (MCC), balanced accuracy score (BACC), F1 score, the area under the receiver operating characteristic curve (AUROC), the area under the precision–recall curve (AUPRC) and specificity.
(1)$$accuracy=\frac{TP+TN}{TP+TN+FP+FN}$$
(2)$$recallrate=\frac{TP}{TP+FN}$$
(3)$$precision=\frac{TP}{TP+FP}$$
(4)$$MCC=\frac{TP\times TN-FP\times FN}{\sqrt{\left(TP+FP)(TP+FN)(TN+FP)(TN+FN\right)}}$$
(5)$$BACC=\frac{\left(TPR+TNR\right)}{2}$$
(6)$$F1score=\frac{2\times (precision\times recallrate)}{precision+recallrate}$$
(7)$$AUROC=\int_{x=0}^{1}TPR\left({FPR}^{-1}(x)\right)dx$$
(8)$$AUPRC=\int_{-\infty }^{+\infty }precision\left(x\right)dP\left[Y\leq x\right]$$
(9)$$specificity=\frac{TN}{TN+FP}$$

## 5. Conclusions

In conclusion, our study underscores the potential of deep chemical language models to predict ADRs induced by marketed drugs. Our MoLFormer-XL-CNN model outperformed conventional QSAR approaches across multiple datasets, showcasing its efficacy in ADR prediction with enhanced structural representation and predictive accuracy. These findings emphasize the transformative impact of deep chemical language models, not only in pharmacovigilance but also in facilitating early ADR detection during drug development.

## Figures and Tables

**Figure 1 ijms-25-04516-f001:**
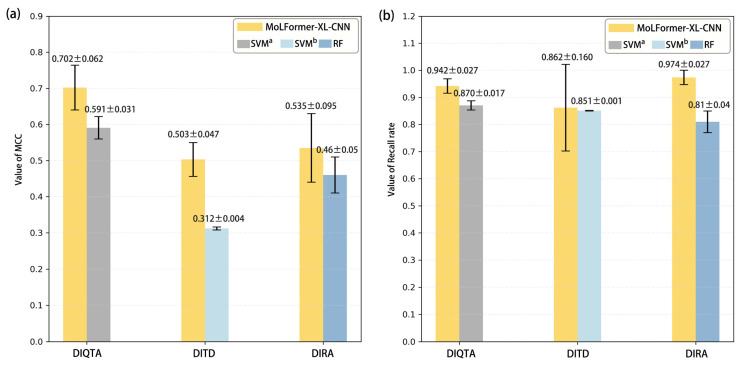
The comparison of the predictive results with three previous studies. SVM^a^: Wulin Long et al. established an SVM model to predict DIQT based on the DIQTA dataset [13]. SVM^b^: Liyuan Kang et al. established an SVM model to predict DIT based on the DITD dataset [12]. RF: Yifan Zhou et al. established an RF model to predict DIR based on the DIRA dataset [16]. (**a**) The comparison of the value of MCC. (**b**) Comparison of the value of recall rate.

**Figure 2 ijms-25-04516-f002:**
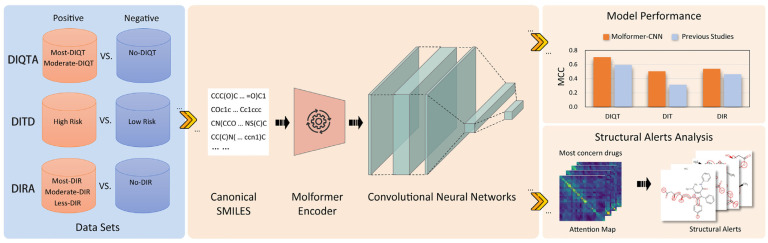
Workflow of our study.

**Table 1 ijms-25-04516-t001:** Prediction results of MoLFormer-XL-CNN model for three datasets.

	DIQTA	DITD	DIRA
Accuracy	0.859 ± 0.029	0.750 ± 0.037	0.866 ± 0.034
Recall rate	0.942 ± 0.027	0.862 ± 0.160	0.974 ± 0.027
Precision	0.845 ± 0.029	0.771 ± 0.010	0.872 ± 0.029
MCC	0.702 ± 0.062	0.503 ± 0.047	0.535 ± 0.095
BACC	0.835 ± 0.033	0.719 ± 0.058	0.703 ± 0.042
F1 score	0.891 ± 0.022	0.800 ± 0.052	0.920 ± 0.023
AUROC	0.829 ± 0.060	0.702 ± 0.048	0.703 ± 0.042
AUPRC	0.822 ± 0.079	0.742 ± 0.101	0.832 ± 0.062
Specificity	0.747 ± 0.036	0.576 ± 0.260	0.432 ± 0.088

**Table 2 ijms-25-04516-t002:** Attention analysis for drugs with a high risk of DIQT.

Generic/ Proper Name(s)	Canonical SMILES	Structures Extracted by Attention Map	SAs*
Quinidine gluconate	C=CC1CN2CCC1CC2C(O)c1ccnc2ccc(OC)cc12	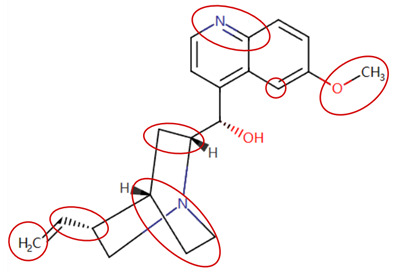	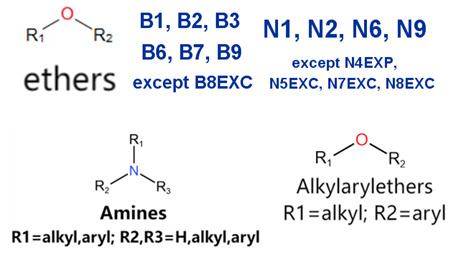
Vandetanib	COc1cc2c(Nc3ccc(Br)cc3F)ncnc2cc1OCC1CCN(C)CC1	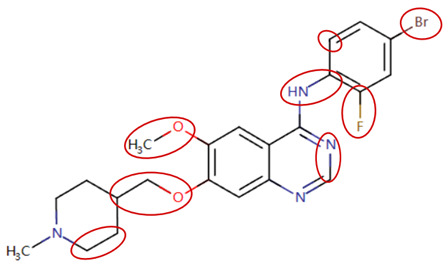	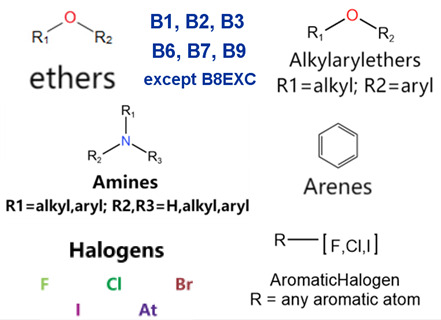
Ibutilide fumarate	CCCCCCCN(CC)CCCC(O)c1ccc(NS(C)(=O)=O)cc1	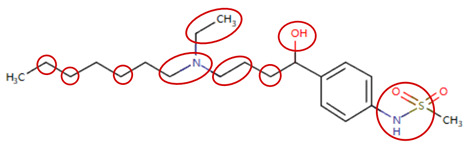	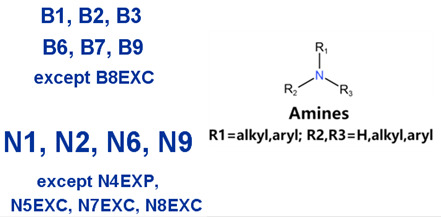
Dofetilide	CN(CCOc1ccc(NS(C)(=O)=O)cc1)CCc1ccc(NS(C)(=O)=O)cc1	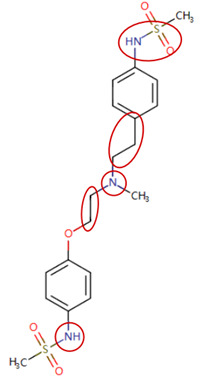	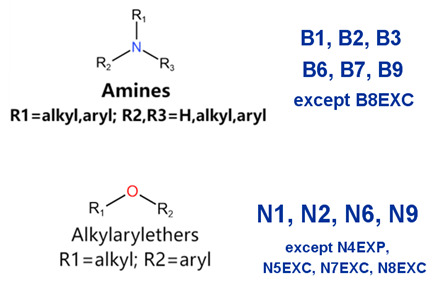
Disopyramide phosphate	CC(C)N(CCC(C(N)=O)(c1ccccc1)c1ccccn1)C(C)C	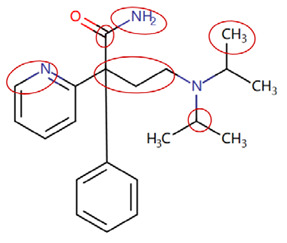	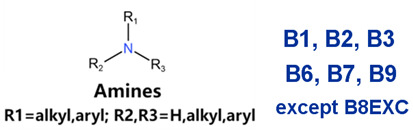

SAs*: SAs obtained from the ToxAlert platform in the Online Chemical Database (OCHEM, https://ochem.eu/home/show.do accessed on 6 February 2024).

**Table 3 ijms-25-04516-t003:** Attention analysis for antiepileptic drugs with a high risk of DIT.

Generic/ Proper Name(s)	Canonical SMILES	Structures Extracted by Attention Map	SAs*
Phenytoin	C1=CC=C(C=C1)C2(C(=O)NC(=O)N2)C3=CC=CC=C3	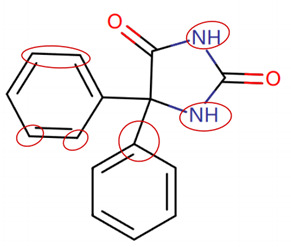	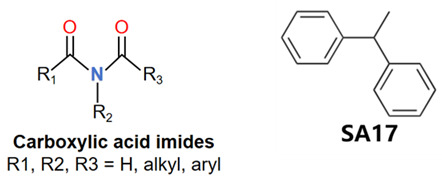
Valproate	CCCC(CCC)C(=O)O	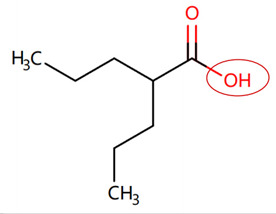	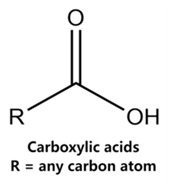

SAs*: SAs obtained from the ToxAlert platform in the Online Chemical Database (OCHEM, https://ochem.eu/home/show.do accessed on 6 February 2024).

**Table 4 ijms-25-04516-t004:** Attention analysis for statin.

Generic/ Proper Name(s)	Canonical SMILES	Structures Extracted by Attention Map	SAs*
Fluvastatin sodium	CC(C)n1c(C=CC(O)CC(O)CC(=O)O)c(-c2ccc(F)cc2)c2ccccc21	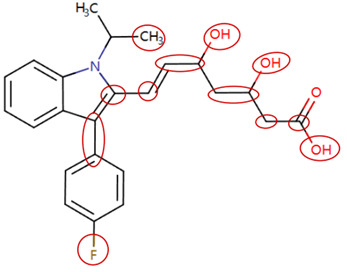	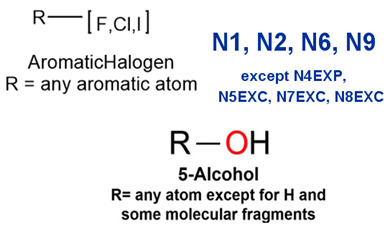
Lovastatin	CCC(C)C(=O)OC1CC(C)C=C2C=CC(C)C(CCC3CC(O)CC(=O)O3)C21	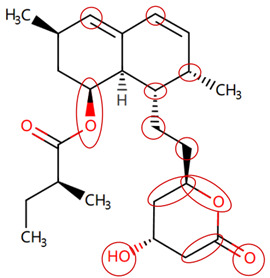	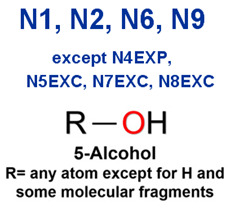
Pravastatin sodium	CCC(C)C(=O)OC1CC(O)C=C2C=CC(C)C(CCC(O)CC(O)CC(=O)O)C21	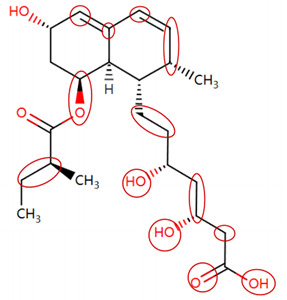	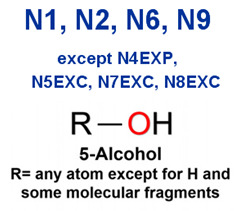
Atorvastatin calcium	CC(C)c1c(C(=O)Nc2ccccc2)c(-c2ccccc2)c(-c2ccc(F)cc2)n1CCC(O)CC(O)CC(=O)O	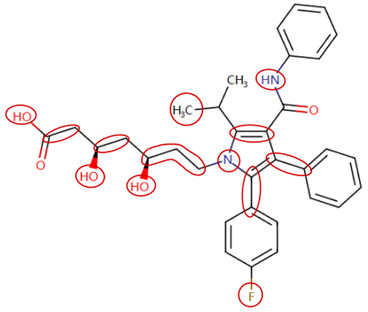	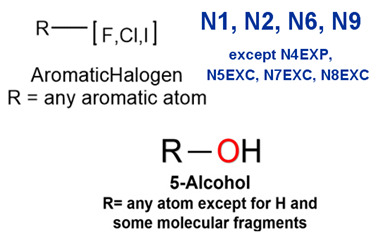
Rosuvastatin calcium	CC(C)c1nc(N(C)S(C)(=O)=O)nc(-c2ccc(F)cc2)c1C=CC(O)CC(O)CC(=O)O	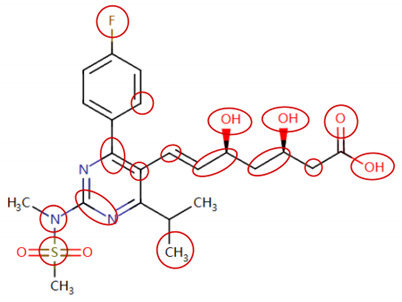	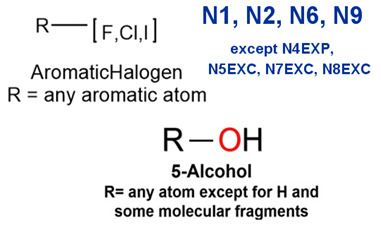
Pitavastatin calcium	O=C(O)CC(O)CC(O)C=Cc1c(C2CC2)nc2ccccc2c1-c1ccc(F)cc1	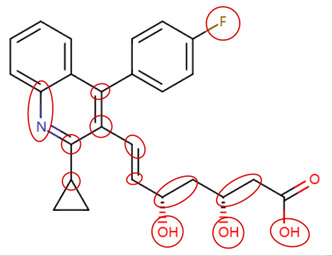	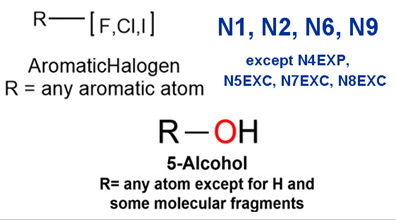
Simvastatin	CCC(C)(C)C(=O)OC1CC(C)C=C2C=CC(C)C(CCC3CC(O)CC(=O)O3)C21	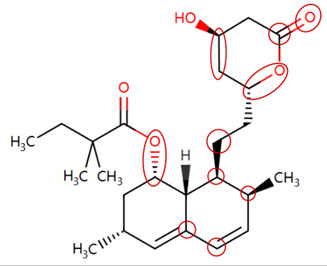	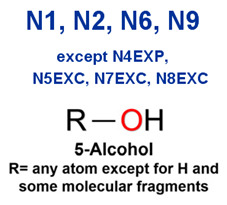

SAs*: SAs obtained from the ToxAlert platform in the Online Chemical Database (OCHEM, https://ochem.eu/home/show.do accessed on 6 February 2024).

## Data Availability

The data presented in this study are openly available at https://github.com/LiSH7450/MoLFormer-XL-CNN_model accessed on 6 February 2024.

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
