# Peer review of "Advancing Adverse Drug Reaction Prediction with Deep Chemical Language Model for Drug Safety Evaluation"

_ijms, 2024, doi:10.3390/ijms25084516_

Round 1

Reviewer 1 Report

Comments and Suggestions for Authors

Overview

The current manuscript entitled “Advancing Adverse Drug Reaction Prediction with Deep Chemical Language Model for Drug Safety Evaluation” leverages the pre-trained chemical language model MoLFormer-XL jointly with convolutional neural networks (CNNs) and has demonstrated advancements over traditional machine learning approaches in predicting ADRs. Also, the model identifies key molecular features associated with ADR risks via attention analysis, which offers new insights into structural alerts for drug safety assessment.

The objectives and the rationale of the study are clearly stated. The statistical analyses are well reported. The interpretation of results and study conclusions supported by the data.

General questions and comments

The information flow in the introduction may be improved for better readability:

1.     Consider elaborating more on the Structure-Activity Relationship (SAR) methodologies to better inform reader of the relevance and necessity of such approaches.

2.     Line 48,49 seem to appear earlier than expected since the authors have not yet comment on conventional machine learning models.

3.     Line 78,79, this is a bit ambiguous, where it is not clear whether the CNN is already part of the MoLFormer-XL model.

Considering adding more brief background information about the three datasets used in the study (DIQTA, DITD and DIRA), e.g., what were measured for each drug in these datasets, how was it measured, what data structure was used to present those records, etc.

While the authors clearly emphasized the strengths of the study, more statements about the limitations of the current study are expected.

Specific questions and comments

Figure 1 is ambiguous, what studies are “previous studies” referring to? It seems the authors are using a single bar to represent the MCC & recall rate of “previous studies”? What are these values, are they the mean of “previous studies”? And the figure captions are not informative.

Line 246 – 250: the corresponding URL for DITA seems to be missing in the texts. Also, the download URL for DIQTA and DIRA data fails to fetch the webpage when I test them (there may an extra slash by the end of the URL). Please double check.

Author Response

We thank the reviewer for recognizing the value of our study, and for your valuable feedback and suggestions. We have made corresponding modifications to improve our research based on your suggestions.

  1. General questions and comments

The information flow in the introduction may be improved for better readability:

(1). Consider elaborating more on the Structure-Activity Relationship (SAR) methodologies to better inform reader of the relevance and necessity of such approaches.

A: Thanks for this suggestion. We have further elaborated on the Structure-Activity Relationship (SAR) methodologies to better demonstrate the relevance and necessity of such approaches (Line 36-42).

(2). Line 48,49 seem to appear earlier than expected since the authors have not yet comment on conventional machine learning models.

A: Thanks to the reviewer for raising this question. We have adjusted the wording of the relevant paragraphs and mentioned in conventional machine learning models earlier, making the research methods in subsequent paragraphs more reasonable.

(3). Line 78,79, this is a bit ambiguous, where it is not clear whether the CNN is already part of the MoLFormer-XL model.

A: Thanks to the reviewer for raising this question. We revised the sentence and make it clear that CNN is not a part of the MoLFormer-XL model.

(4). Considering adding more brief background information about the three datasets used in the study (DIQTA, DITD and DIRA), e.g., what were measured for each drug in these datasets, how was it measured, what data structure was used to present those records, etc.

A: Thanks for this suggestion. We adding a brief description of the methodology used to categorize drug risk in the three datasets used in the study (DIQTA, DITD and DIRA).

(5). While the authors clearly emphasized the strengths of the study, more statements about the limitations of the current study are expected.

A: Thanks for this suggestion. We have refined statements made appropriate additions about the limitations of the current study are mentioned in the last paragraph of the Discussion.

  1. Specific questions and comments

(1). Figure 1 is ambiguous, what studies are “previous studies” referring to? It seems the authors are using a single bar to represent the MCC & recall rate of “previous studies”? What are these values, are they the mean of “previous studies”? And the figure captions are not informative.

A: Thanks to the reviewer for raising this question. The previous studies were three studies that all leveraged conventional machine learning algorithms in conjunction with molecular descriptors, and used the same datasets to predict DIQT,DIT,DIR respectively. A quick summary of previous studies in Figure 1 was added in Line 114-121.

In Figure 1, the bars were the result of these three studies, the labeling of the bars was changed to indicate the methods used in previous studies for each dataset, and a description of previous studies was added to the Figure legend.

(2). Line 246-250: the corresponding URL for DITA seems to be missing in the texts. Also, the download URL for DIQTA and DIRA data fails to fetch the webpage when I test them (there may an extra slash by the end of the URL). Please double check.

A: Thanks to the reviewer for raising this question. Line 246-250, we checked and corrected the URL links to the DIQTA dataset and DIRA dataset, and added the URL link for the DITD dataset.

Thank you for your valuable feedback, and we hope that these changes have addressed your concerns. If you have any further questions or comments, please do not hesitate to contact us. The revised manuscript please see the attachment.

Reviewer 2 Report

Comments and Suggestions for Authors

I found this paper extremely interesting. I have been looking at LLMs for other areas of science and to apply it to SMILES strings was new to me. The paper is very well written and explained. It suffers a little from paucity of data as the authors acknowledge ADRs are a rare event - idiosyncratic! However, the stats and the recognition of false positives make me think that even if I won't blindly trust the predictive power of the model yet it provides an interesting angle that may help the bigger picture.
The only quibble I had was for Figure 1 a quick summary to remind what the state-of-the-art methods from previous studies being compared with actually were.

Author Response

We thank the reviewer for recognizing the value of our study, and for your valuable feedback and suggestions. We have made corresponding modifications to improve our research based on your suggestions.

  1. The only quibble I had was for Figure 1 a quick summary to remind what the state-of-the-art methods from previous studies being compared with actually were.

A: Thanks for this suggestion. In Figure 1, the labeling of the bars was changed to indicate the methods used in previous studies for each dataset, and a description of previous studies was added to the Figure legend. A quick summary of previous studies in Figure 1 was added in Line 114-121.

Once again, thank you for your attention and support of our research. If you have any further questions or suggestions, please do not hesitate to contact us. The revised manuscript please see the attachment.

Reviewer 3 Report

Comments and Suggestions for Authors

In recent years, deep chemical learning has become important in areas of interest in artificial intelligence.

The authors proposed the MoL.Former-XL model, in combination with convolutional neural networks, for recognising complex relationships between the molecular structure of drugs and the resulting potential for adverse effects.

The work is very interesting. The new tool proposed by the authors could, in the future, make it possible, at the stage of drug product development, to generate drug candidates after analysing their molecular structure and identifying chemical substructures associated with adverse effects.

My comments

- The Introduction chapter should contain the aims of the work, not the findings, which can be referred to in subsequent chapters of the manuscript, including, above all, the conclusion.

- Authors should redact the literature of the References section in accordance with editorial requirements.

Author Response

We thank the reviewer for recognizing the value of our study, and for your valuable feedback and suggestions. We have made corresponding modifications to improve our research based on your suggestions.

  1. The Introduction chapter should contain the aims of the work, not the findings, which can be referred to in subsequent chapters of the manuscript, including, above all, the conclusion.

A: Thanks for this suggestion. We have revised the last paragraph of the Introduction section to be more statements about the aims of our work and less descriptive of our findings.

  1. Authors should redact the literature of the References section in accordance with editorial requirements.

A: Thanks for this suggestion. We have checked and redacted the literature of the References section in accordance with editorial requirements.

Thank you for your valuable feedback, and we hope that these changes have addressed your concerns. If you have any further questions or comments, please do not hesitate to contact us. The revised manuscript please see the attachment.